# Technical Note: Investigating the potential for smartphone-based monitoring of evapotranspiration and land surface energy balance partitioning

Adriaan J. Teuling[1], Belle Holthuis[1] & Jasper F. D. Lammers[1]

[1]Hydrology and Environmental Hydraulics Groups, Wageningen University & Research, Wageningen, 6700AA, Netherlands

*Correspondence to*: Adriaan J. Teuling (ryan.teuling@wur.nl)

**Abstract.** Evapotranspiration plays a key role in the terrestrial water cycle, climate extremes and vegetation functioning. However, the understanding of spatio-temporal variability of evapotranspiration is limited by a lack of measurement techniques that are low-cost, and that can be applied anywhere at any time. Here we investigate the estimation of evapotranspiration and land surface energy balance partitioning using only observations made by smartphone sensors. Individual variables known to effect evapotranspiration as measured by smartphone sensors generally showed a high correlation with routine observations during a multi-day field test. In combination with a simple multivariate regression model fitted on observed evapotranspiration, the smartphone-observations had a mean RMSE of 0.10 and 0.05 mm/h during validation against lysimeter and eddy covariance observations, respectively. This is comparable to an error of 0.08 mm/h that is associated with estimating the eddy covariance ET from the lysimeter or vice versa. The results suggests that smartphone-based ET monitoring could provide a realistic and low-cost alternative for real-time ET estimation in the field.

## 1 Introduction

In most climates, more rainfall returns to the atmosphere via evapotranspiration than ends up in rivers. Evapotranspiration (commonly referred to as ET) also modulates near-surface climate by limiting the amount of direct warming by sensible heat fluxes. Under conditions of low soil moisture, reduced ET reflects ecosystem water stress, reduced carbon uptake, and a loss of agricultural production, as well as enhanced atmospheric warming through a shift in the land surface energy balance reflected in enhanced land surface temperatures. This makes ET a key indicator of environmental conditions and global change (Seneviratne et al., 2010; Denissen et al., 2022). In spite of its importance, few, if any, government agencies are tasked with the routine monitoring of ET. In addition, important gaps exist in our current ability to monitor ET, in particular limiting our understanding of how ET interacts with droughts and heatwaves (Teuling et al., 2013; Miralles et al., 2019; Lansu et al., 2020). Enhanced ET observation is key to filling those gaps.

Traditionally, ET has been measured through the mass-balance principle applied to catchments or lysimeters. While this approach is generally accurate (Allen et al., 2011, Senay et al., 2011), it provides limited spatial and/or temporal detail. Flux towers equipped with eddy covariance sensors also measure ET through turbulent moisture transport, but such sites are

expensive to maintain, current tower locations are typically chosen for their relevance to the carbon balance (e.g. bias towards wetter sites with high carbon uptake) rather than to soil moisture-temperature coupling, and the footprint varies with wind conditions. ET can alternatively be estimated from Earth Observation (EO), typically using the thermal infrared atmospheric window of the electromagnetic spectrum (Derardja et al., 2024). While such approaches give valuable insight into the spatial distribution of ET, they rely on available satellite overpasses, cloud-free conditions, and ET inference models (Amani and

Shafizadeh-Moghadam, 2023). Most ET inference models and more classical potential ET-based methods have been developed in times when actual ET observations were scarce. Due to increasing availability of observations in hydrology, but also ET in particular, machine learning approaches now often outperform existing models due to their ability to optimally utilize the information in observations (Kratzert et al., 2019). This calls for development of new observation methods to close the observational blind spot – methods that are low-cost, flexible, operating in real-time at high spatial and temporal resolution,

and making use of machine learning where appropriate.

Over the past decade, application of mobile phone technology to measure the terrestrial part of the hydrological cycle and associated meteorological variables has been gaining traction. It has been shown that precipitation, for instance, can be estimated from microwave links used in commercial cellular communication networks (Messer et al., 2006; Overeem et al., 2013). Several free and commercial apps exist that can be used to monitor river discharge often based on water level and/or

surface velocity estimates (Kampf et al., 2018; Fehri et al., 2020, Damtie et al., 2023). Air temperature can be estimated from sensors that monitor phone battery temperatures (Overeem et al., 2013), incoming radiation can be estimated from a calibrated phone's light sensor (Al-Taani and Arabasi, 2018, Hukseflux, 2023), while external sensors have been developed for wind speed, temperature, pressure and humidity normally provided by weather stations. However all these estimates based on mobile phone technology would at best complement routine estimates of temperature, precipitation, or discharge made by dedicated

government agencies. Measuring ET directly by smartphone has remained elusive.

Ongoing advances in sensor developments now provide new opportunities. In particular, thermal infrared imagers have become more compact and affordable, allowing them to be integrated in a smartphone. In combination with other build-in or external handheld sensors for relevant meteorological variables, this allows for direct inference of evapotranspiration through the land surface energy balance. This procedure is conceptually similar to evapotranspiration estimation from Earth observation, but

with the added benefits that it can be done in real-time, based on local meteorological conditions, and independent of cloud cover and satellite overpasses. This setup can be used to measure temporal evolution of surface energy balance partitioning at a specific location, or spatial patterns of flux partitioning, in particular in areas with high spatial variability such as urban environments.

While smartphones can potentially monitor all variables relevant for ET, the question is if these estimates, when combined ,

provide enough information for accurate ET estimation under field conditions. Therefore, the primary goal of this feasibility study is to investigate how well smartphone-based estimates of surface fluxes from measurements of individual meteorological variables validate against routine measurements made by lysimeters and eddy covariance. To this end, two main research questions are addressed, namely: 1) Do handheld sensors provide robust estimates of standard meteorological variables

relevant for ET estimation? and 2) Can a simple multivariate regression model fitted to smartphone observations provide accurate ET estimates? These questions are addressed using observations made during field testing at a measurement site equipped with standard meteorological instrumentation, a large weighing lysimeter, and an eddy covariance tower to allow for validation of the individual meteorological variables as well as flux estimates.

## 2 Methods and Data

Figure 1 illustrates how smartphone-based ET monitoring might look in practice. In this study, a smartphone (model CAT S62 Pro, referred to as S62 from hereon) was used to record surface temperature ($T_s$) using its build-in FLIR Lepton 3.5 thermal sensor. Because we focus on vegetated conditions, we assume emissivity does not differ from unity. Global radiation was estimated using the S62's build-in light sensor, where the sensor was covered with 2 layers of standard paper in order to avoid sensor saturation when the sensor is exposed to direct sunlight. This procedure is similar to Hukseflux (2023). Because a phone's lens typically does not capture light from all angles equally, luminance ($I$) measurements were taken with the phone held straight-up perpendicular to the sun, and the readings were later corrected for the solar angle using the phone's pitch ($\phi$, the angle between a plane parallel to the device's screen and a plane parallel to the ground). Both luminance from the light sensor and pitch were recorded using the Sensors app. A WeatherFlow WEATHERmeter, connected to the S62 via Bluetooth, was used to simultaneously record air temperature ($T_a$), pressure, relative humidity (RH), and wind speed ($w_s$). To prevent bias, the WEATHERmeter was kept in a shaded and ventilated place in between measurements. The measurement principle is illustrated in Figure 1.

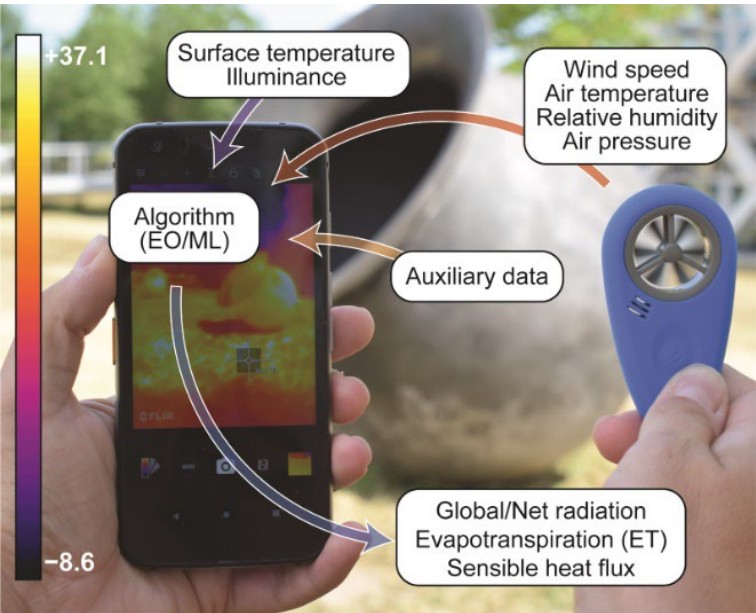

**Figure 1: Principle of smartphone-based monitoring. Due to the cooling effect of evapotranspiration, surface temperature reflects the partitioning of global radiation into evapotranspiration and sensible heat. Both the global radiation and the surface temperature**

85 can be measured by phone's internal sensors, while an external sensor can provide meteorological variables that affect evapotranspiration and surface energy balance partitioning. Colour bar indicates surface temperature (°C) as seen on screen. Picture taken on 5/8/22 by Janneke Remmers on Wageningen campus amid the 2022 summer drought (ambient air temperature 21°C).

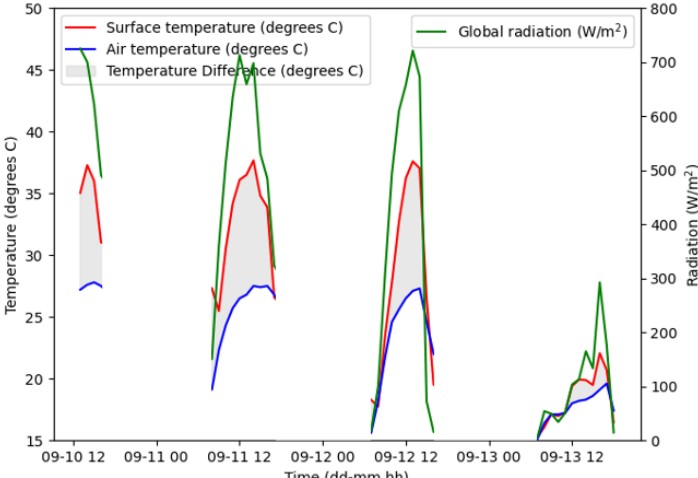

**Figure 2: Overview of conditions during Rietholzbach field campaign. Smartphone temperature observations ($T_s$ and $T_a$) and Buël global radiation during the field campaign. Over the course of the 4-day campaign, measurements were taken at 36 moments.**

The field data were collected during daytime under rainless conditions from 10–13 September 2023 at the Büel meteorological station (Gähwil, Sankt Gallen, Switzerland), which is located within the pre-alpine Rietholzbach catchment. Data from this site has been used for numerous hydro(meteoro)logical studies (Teuling et al., 2010; Seneviratne et al., 2012; Hirschi et al., 2017; Michel & Seneviratne, 2022). The site was chosen because ET is measured independently by a large weighing lysimeter (area 3.14 m² and depth 2.5 m) and eddy covariance. Hourly values for standard meteorological variables, eddy covariance fluxes of sensible and latent heat, and lysimeter evapotranspiration were used to complement and validate the smartphone observations. The smartphone and Büel observations are available from Teuling (2024). An overview of the conditions during the data collection is given in Figure 2, revealing a wide range of temperature and radiation conditions. It also illustrates the temporal dynamics of the difference between surface and air temperature, which is largest near the daily global radiation peak reflecting the strongest turbulent heat fluxes. Over the course of the 4-day campaign, measurements were taken at 36 moments. During the field campaign, estimating one direct ET observation (lysimeter or eddy covariance) by the other resulted in an RMSE of 0.084 mm/h, which for this site can be seen as a practical upper limit for errors associated with ET estimation in these conditions since it reflects the inherent uncertainty between two state-of-the-art methods. The sum of latent and sensible heat fluxes over the field campaign explained 98.7% of the net radiation, so a lack of energy balance closure likely does not explain this uncertainty.

From both theoretical considerations and observational evidence it is known that ET depends on a range of environmental variables. It is often assumed that these dependencies take a linear form, and many field studies have confirmed the validity of this assumption for different systems (Maes et al., 2019, Lansu et al., 2020, Jansen et al., 2022). For our initial testing, and given the limited amount of data available for this study, we use the following simple multivariate regression instead of a more complex machine learning algorithm to estimate the instantaneous evapotranspiration:

$$ET_{phone} = \alpha_I \times I \times \cos\phi + \alpha_{RH} \times RH + \alpha_{Ta} \times T_a + \alpha_{Ts} \times T_s + \alpha_{ws} \times w_s + c \tag{1}$$

This model calculates the individual contribution of each variable to the total observed ET after calibration of the $\alpha$ coefficients and intercept $c$.. E.g., when $\alpha_I > 0$, the illuminance will add to the ET budget. If the coefficient is negative, the component (coefficient multiplied with variable) will be subtracted from the ET budget. For the local application and validation in this study, the coefficients are calibrated such that the calculated ET resembles observed ET best. For a more general application, more complex ML models trained with a more extensive dataset or integration of EO models with smartphone data should be considered, though it should be noted that the performance of EO models for ET estimations is not necessarily high when evaluated at smaller local scales that are the focus of this study (e.g. Pardo et al., 2014, Cheng et al., 2021). From the collected data, two thirds were randomly selected to calibrate the regression models for $ET_{phone}$, while the remainder were used to validate the obtained model. For $ET_{phone}$, this procedure was repeated 2000 times, and validation error statistics were calculated as the mean over the resulting sample. For many practical applications, the interest would be on daily rather than instantaneous (or hourly) ET values. Various upscaling methods are available from the Earth observation literature to do this (see Jiang et al., 2021), but these are not used in this study since our method is not limited to available satellite overpasses and multiple observations can be taken for a more robust estimation of the daily mean.

## 3. Results

Instantaneous observations from individual variables by smartphone sensors generally showed a good correlation with hourly values recorded at Buël. Air temperature ($R^2 = 0.88$), relative humidity ($R^2 = 0.80$) and air pressure ($R^2 = 0.98$) showed the highest correlations. Wind speed showed a satisfactory correlation ($R^2 = 0.57$), likely because of its higher temporal variability, the low wind conditions during the field campaign, and the discrepancy between the instantaneous smartphone-based observations and the hourly average values at Buël. Global radiation could not be measured directly, but instead a linear model for its estimation was calibrated on the subset of the pitch-corrected illuminance values. Validation on the remaining part of the data revealed a high correlation (validation $R^2 = 0.97$, see Figure 3a). Besides information on meteorological conditions and energy driving the land-atmosphere exchange, it is clear that the measurements also reflect key land-atmosphere exchange processes. This is illustrated by the high correlation between the smartphone surface-air temperature difference and the observed sensible heat flux (validation $R^2 = 0.90$, see Figure 3b).

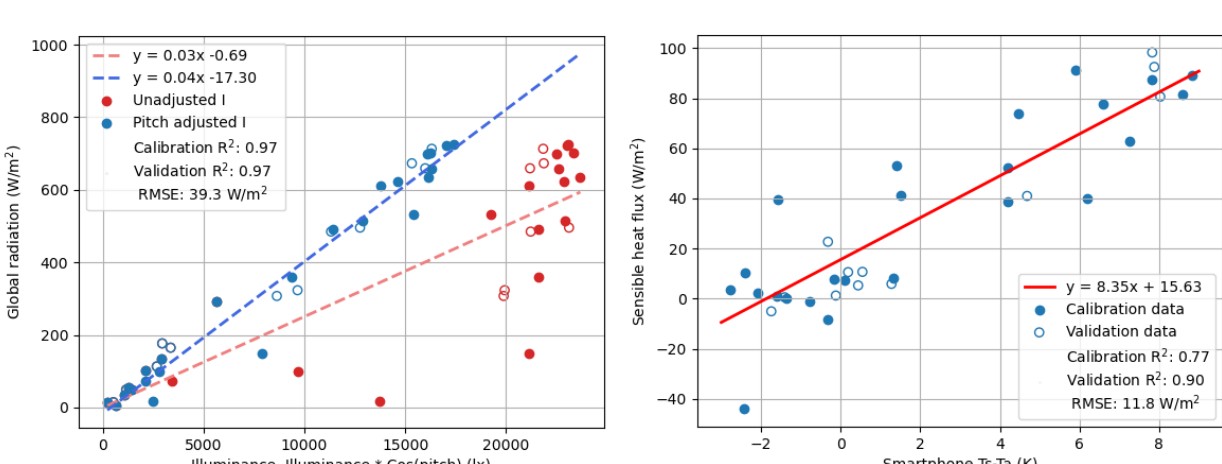

**a** **b**

**Figure 3: Smartphone monitoring and calibration of radiation and heat fluxes. a) Impact of pitch-adjustment on estimation of global radiation from smartphone-measured illuminance. b) Relation between sensible heat flux from eddy covariance and smartphone-measured $T_s-T_a$.**

In a next step, we estimate the evapotranspiration as observed by lysimeter and eddy covariance by fitting Eq. 1 solely with smartphone observations. Validation of this model reveals a good performance, with relatively small mean RMSE values of 0.102 mm/h (lysimeter) and 0.050 mm/h (eddy covariance) across the 2000-member ensembles. Figure 4a illustrates the performance for ensemble members that are representative for the mean performance. These values present the expected error of the proposed smartphone method when fitted on a small site-specific dataset. Interestingly, the errors are considerably smaller (eddy covariance) and only slightly larger (lysimeter) in comparison to the uncertainty arising from a direct comparison between the two state-of-the-art methods (Figure 4b). This suggests that even with limited site-specific calibration, the method might perform as well as other standard methods.

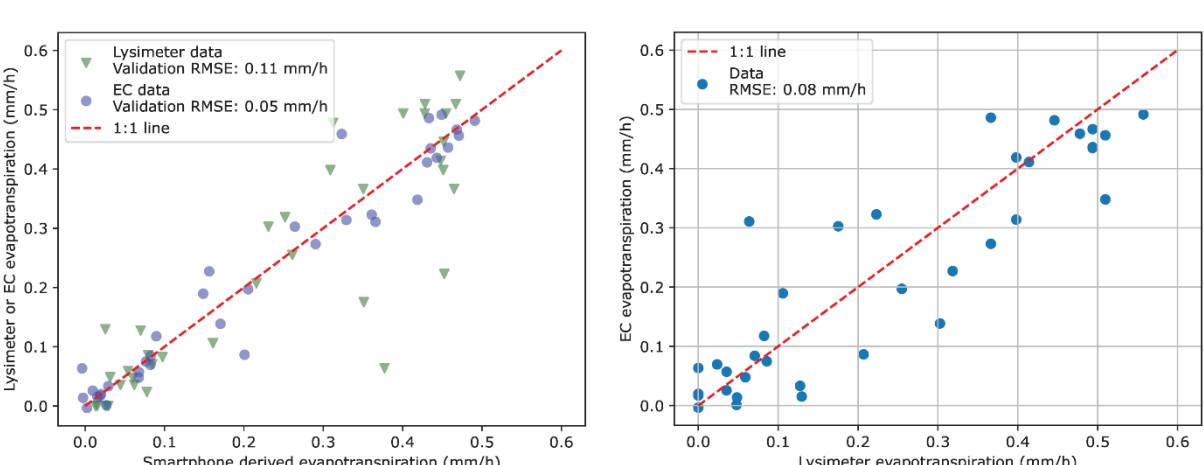

Figure 4: Illustration of ET prediction performance. a) Illustration of ET_phone vs ET observed by lysimeter and eddy covariance. The 2 models used for ET_phone (for lysimeter and eddy covariance) were each selected out of a 2000-member ensemble because of their RMSE values being close to the mean across all sets, and thus represent an average model outcome. Shown RMSE values are for validation points only, whereas the graph shows all data. b) Relation between ET as observed by lysimeter and eddy covariance as reference.

In the MLR model, it was found that most observations contributed information to ET_phone (Figure 5). Surface temperature, air temperature, and relative humidity were found to contribute most information, while wind speed was found to play a negligible role. It should be noted that wind was generally light during the field campaign, which might explain its small contribution. In spite of the site being well-known for having an energy-limited evapotranspiration regime (Teuling et al., 2013, Michel & Seneviratne, 2022), and global or net radiation generally being a sufficient sole predictor for daily ET under these conditions (Maes et al., 2019), the illuminance term (as a proxy for global radiation) on average contributed less to the ET budget than temperature and humidity. This can be explained by the strong cross-correlations between states at or near the land surface and radiation, in particular at hourly timescales, combined with the relatively short length of the calibration data. It should be noted that the magnitude of the offset term (intercept $c$) is directly related to the units used for the variables in combination with the linearity of their relation to ET. In addition, the relative contribution of the different terms, in particular the temperature and RH terms, showed considerable spread. Nonetheless, this analysis shows that hourly ET estimation benefits from having observations of all relevant variables.

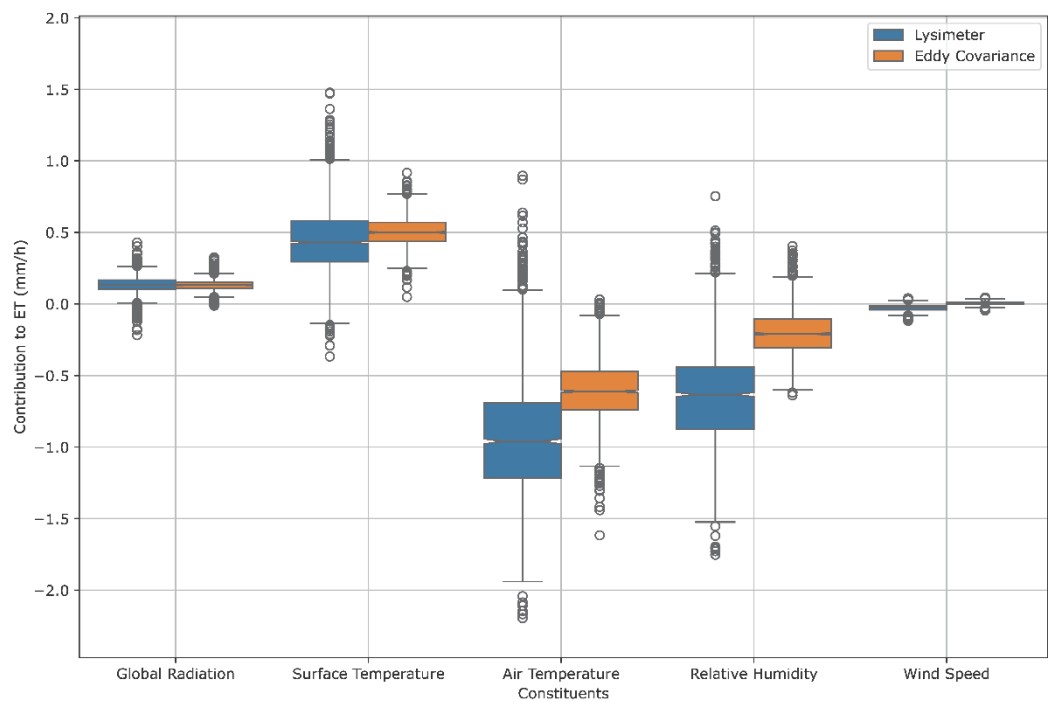

**Figure 5: Distribution of the contributions to the ET budget in the MLR model fitted to subsets of the Büel data. Each distribution contains 2000 values (see Methods) of the matching terms in Eq. 1. Note that the distribution of values for the intercept *c* is not shown.**

## 4. Discussion and Outlook

In this research, we presented the first results of a feasibility study aimed at monitoring evapotranspiration solely using smartphone-based sensors. Based on observations made during a short field campaign at a well-instrumented site in the Swiss pre-alps, we conclude that most meteorological variables relevant to ET estimation are monitored with good to sufficient accuracy by smartphone sensors. When a simple machine learning algorithm is fitted on a subset of the observations, validation on independent lysimeter and eddy covariance observations shows mean RMSE values in the range of 0.05–0.11 mm/h. This is comparable to the difference between these two state-of-the-art techniques during the field campaign (RMSE 0.08 mm/h), and similar to errors found in comparison between large-scale estimates and eddy covariance (RMSE 0.04–0.14 with median 0.07 mm/h, see Bayat et al., 2024). Analysis of the machine learning algorithm outputs showed that for this short feasibility study, observations of radiation, temperature (both surface and air) and humidity all provided information, but wind less so. While these results show that smartphone ET estimation can give accurate values after local calibration, they do not provide information on the performance at other sites where no calibration data is available.

In order to investigate the transferability of the method to other sites, a second measurement campaign was conducted on 2
days in April 2024 at the TERENO lysimeters located in the Rollesbroich hydrological observatory (Qu et al., 2016). Direct
application of the model (Eq. 1) calibrated to subsets of the Büel observations as described earlier gave a satisfactory model
performance with a median RMSE of around 0.10 mm/h for each of the six lysimeters. This performance however increased
considerably after local calibration following the same procedure as used earlier for Büel, with median RMSE values in the
range 0.06–0.07 mm/h. This shows that the general methodology works at different sites but best results are obtained after
calibration. A closer look into the difference between the models calibrated on Büel vs. Rollesbroich data (Figure 6) provides
an explanation for the poorer model performance. Besides warmer temperatures encountered at Büel, wind speeds were lower
during the Rietholzbach campaign (order 0.5–1 m/s) than during the Rollesbroich campaign (4–5 m/s). As a result, the gradient
between surface and air temperature was much smaller at Rollesbroich, and wind becomes a more important predictor in the
model (Figure 6) at the expense of temperature (see the difference between the these constituents in Figures 5 and 6). This
shows that for future application, a more complex model that is trained on a more complete range of weather conditions is
needed.

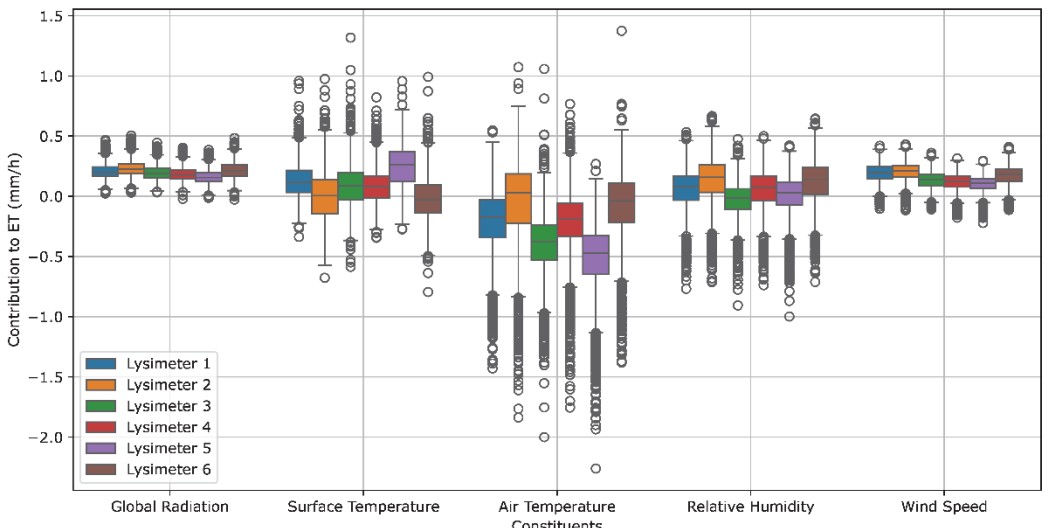

**Figure 6: Distribution of the contributions to the ET budget in the MLR model fitted to subsets of the Rollesbroich campaign data.**
**Each distribution contains 2000 values (see Methods) and is shown separately for each of the six lysimeters. Note the contribution**
**of wind which is nearly absent in the Rietholzbach case (Fig. 5).**

The technology used in this study can be considered low cost at a current price tag of around 750 EUR/USD (650 for phone
and 90 for WEATHERmeter). Most common smartphones can be equipped with an external thermal camera for around 230

EUR/USD. It should be noted that the sensors in these phones, in particular the light sensor and its lens, have not been optimized for the current application. Further future improvements should thus be possible. This is also true for the algorithm. The flux data from Büel used here reflects humid conditions over grassland as evidenced by a Bowen ratio of 0.22 based on average fluxes during the field campaign. In the future, a more complex machine learning algorithm should be trained with more data from a range of climatological, geographical, and land cover conditions. The current study was designed as a

feasibility study, where ET was estimated in hindsight. Ideally, in future applications a dedicated app would receive input from the various sensors in real-time, and directly infer ET from those using a further optimized algorithm. Such algorithm could for instance also use additional information on albedo (Leeuw and Boss, 2018), time, location, and land cover that was not used in this study. We did not yet investigate how sensitive the results are to the choice for a particular sensor, and how this would affect the need for calibration. This will be the focus of future work. The same principle used here on smartphone data

could potentially also be applied to a combination of a cheap weather station and IR temperature sensor for a more automated monitoring at a single location. However, this would require post-processing on the computer, while a dedicated smartphone app could do the same on the fly.

The prospect to measure evapotranspiration using an affordable, handheld device marks a watershed moment in hydrology. For the first time, hydrologists might be able to measure evapotranspiration anywhere and anytime. We hope this first

feasibility study will lead the community to embrace this opportunity, by developing and calibrating algorithms, possibly aided by the latest generation of precision lysimeters and online data, that will translate the observations into a real-time ET imagery. Such a new data sources would complement current ET monitoring by filling the existing blind-spot, thereby not only helping science but moreover directly supporting operational water management, spatial planning, and irrigation scheduling. With smartphone-based ET monitoring linked to crowdsourcing-based data acquisition, it will be possible to monitor future droughts

and their impacts quickly, and in unprecedented detail.

**Code and data availability**

Smartphone observations and matching observations at Rietholzbach/Büel and Rollesbroich are available at https://www.hydroshare.org/resource/4f88a4b06bc846a1b948d06fe9145223/. The Python scripts used for analysis and creating the figures are available at https://github.com/JasperLammers99/Handheld_Evapotranspiration.

**Author contribution**

AJT developed the initial idea and designed the experiments. JFDL and BH carried out the experiments. JFDL developed the model code and performed the analysis. AJT prepared the manuscript with contributions from JFDL and BH.

**Competing interests**

At least one of the (co-)authors is a member of the editorial board of Hydrology and Earth System Sciences.

**Acknowledgments**

We thank the Land-Climate Dynamics group at the Institute for Atmospheric and Climate Science (IAC), ETH Zurich, in particular Martin Hirschi and Dominik Michel, for providing access to the Büel measurement site and for sharing their data. We thank Marius Schmidt and Jannis Groh for their help with the Rollesbroich data.

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
