# Peer review of "Technical Note: Investigating the potential for smartphone-based monitoring of evapotranspiration and land surface energy balance partitioning"

_EGUsphere, 2023_

## Referee Comment (RC1)

**Major comments**

This scientific quality of this note is overall okay, and its contents are moderately interesting and marginally useful for the kind of research or context presented in the paper (micrometeorological or water balance research). It may be useful for irrigation planning or garden water use monitoring, but I am not at all convinced that it marks a "watershed moment in hydrology".

My reservations relate to the fact that:

- generally instantaneous values of weather variables and ET estimates are less useful than hourly values integrated to e.g. daily values. No operator is going to stand in the field for 24/7 to take these measurements.

- the IR phone images are only useful if one is interested in spatial variability of ET. Perhaps it can be useful for small-scale investigations of ET variability, e.g. in urban settings.

- As I said, no operator is going to stand in the field for 24/7 to take these measurements. Why not buy a cheap weather station, e.g. and Ecowitt one, which is barely more expensive than the WEATHERmeter and supplement it with a cheap surface temperature sensor?

- In fact, how well would the model have done with Ta only and not Ts? Do we really need Ts? Can this be tested and discussed?

- This approach is only useful if there are high-quality ET data (e.g. EC-data) available to calibrate the ML model. For most places and users these are not available, and I am not sure how we would get around that. Unless we used cloud free high-resolution ET estimates to calibrate the ML models for specific settings, where lots of hobbyist weather station data are available.

- What about emissivity? Do the Ts data not need to be corrected for that? And the fact that the measurements are only measured between 8 and 14 micometer(?).

**Specific comments**

Line 24: "Traditionally, ET has been measured through the mass-balance principle applied to catchments or lysimeters". This statement could be expanded a little to help the reader and needs some references.

In fact the whole paragraph between line 24-32 is devoid of any references. This needs fixing.

LIne 29: Replace "thermal infrared window", by "thermal infrared atmospheric window of the electromagnetic spectrum"

Line 58: Say: "model CAT S62 Pro; referred to as S62 from hereon)"

Line 59/60: Explain better what is mean by saturation here, and what causes this? Why these 2 layers of paper, this sounds rather arbitrary.

Line 61: Also, what is meant by "phone held straight-up perpendicular to the sun". You mean that the phone is held vertically? Refer to Fig.1a here? Also, why are you taking a photo of obstacles sitting on the surface, rather than of the actual land surface? I find this confusing. Surely, this angle is only suitable/crucial for the operation of the light sensor, not for the IR image?

Line 64 & 65: use subscripts for $T_a$ and $w_s$ (and for $T_s$ in line 62). You use them later in the equation.

Line 70-71: You say "partitioning of incoming solar radiation into evapotranspiration and sensible heat". It is net radiation that is partitioned into evapotranspiration and sensible heat, but also into soil heat flux. So, this statement is incorrect. Also "incoming solar radiation" is the same as "global radiation". Do you want to stick with one term? The latter one is less intuitive.

Line 71: "Both can be measured by phone's internal sensors.." What does "Both refer to here?

Line 80-81: Can a little bit more information be given here? "The smartphone and Büel observations are available from Teuling and Lammers (2023)". How many measurements/ IR images were taken in the field, and of what kind of surface? Only in the footprint of the EC mast or 'on the lysimeter'?

Line 89: This equation needs a number. Also, the various alphas are not defined properly? Nor is parameter c? Why is pressure not considered? If that is the case ,then take it out of the rest of the paper.

Line 135-136: What is meant with ".. magnitude of the offset term". Is this the parameter c in the equation?

**Technical corrections**
Line 39: It should be "Hukseflux" not "Hukselfux".
Line 85: It should be "...a lack **of** energy balance closure"
Line 130: 'it should be **negligible** role'.
Line 132: It should be "these condition**s**"

---

## Author Comment (AC1)

**Major comments**

This scientific quality of this note is overall okay, and its contents are moderately interesting and marginally useful for the kind of research or context presented in the paper (micrometeorological or water balance research). It may be useful for irrigation planning or garden water use monitoring, but I am not at all convinced that it marks a "watershed moment in hydrology".

My reservations relate to the fact that:

- generally instantaneous values of weather variables and ET estimates are less useful than hourly values integrated to e.g. daily values. No operator is going to stand in the field for 24/7 to take these measurements.

We fully agree with this point. In many practical applications, one would like an estimate of the daily ET sum. The conversion from instantaneous to daily values was however not our focus, since this is a topic that is already well covered in the literature on satellite estimation of ET. These methods are equally valid for our approach. We will acknowledge this in a revised manuscript by including a references to Jiang et al. (2021) in which several upscaling methods are compared. It should be noted that in contrast to satellite estimation of ET, which depends on available overpasses and cloud-free conditions, our method can be used for single or multiple measurements per day. More estimates at various points in the diurnal cycle will always lead to more robust estimates of daily totals.

- the IR phone images are only useful if one is interested in spatial variability of ET. Perhaps it can be useful for small-scale investigations of ET variability, e.g. in urban settings.

Investigation spatial variability of ET in specific settings including urban environments is indeed a logical application of the method. However we believe there is a wider potential because of the low-cost nature of the sensors. Ultimately, we see crowdsourcing of ET at larger (i.e. national) scales as a realistic future outlook.

- As I said, no operator is going to stand in the field for 24/7 to take these measurements. Why not buy a cheap weather station, e.g. and Ecowitt one, which is barely more expensive than the WEATHERmeter and supplement it with a cheap surface temperature sensor?

For local applications at smaller scales over which meteorological conditions are not expected to vary significantly, a cheap weather station could indeed provide an alternative to the WEATHERmeter. When supplemented by surface temperature sensor, the same methodology could in principle be applied to the resulting data. However this would still require post-processing on a separate computer. The advantage of using a smartphone as platform is that with a dedicated app, all the flux properties can be calculated, shown, and used on the fly.

- In fact, how well would the model have done with Ta only and not Ts? Do we really need Ts? Can this be tested and discussed?

Ultimately, both sensible and latent heat fluxes are to a large extent driven by the land-atmosphere temperature gradient. This is illustrated by Fig 2a and Fig 4. Fig 2a shows the relation between the temperature difference Ts-Ta and the sensible heat flux. Since most of the net radiation that is not used for sensible heat is used for ET, this already shows that the temperature difference, and not Ta, is the main driving force of ET. Fig 4 further investigates the importance of the different variables. As can be seen, both Ta and Ts carry significant information on ET, so using only Ta would not result in robust estimates. It should be noted that observations were made in humid (non-water-limited) conditions. Under water-limited conditions, the role of Ts will likely be more important. This will be tested in the future.

- This approach is only useful if there are high-quality ET data (e.g. EC-data) available to calibrate the ML model. For most places and users these are not available, and I am not sure how we would get around that. Unless we used cloud free high-resolution ET estimates to calibrate the ML models for specific settings, where lots of hobbyist weather station data are available.

Based on our results, it is not possible to conclude whether the approach will or will not work at other sites without calibration. At Rietholzbach, we were able to test both the validation (i.e. performance on data not used in the training) for the same site, as well as validation for the lysimeter when trained with eddy covariance data and vice versa. In both cases, we got more than satisfactory results which makes us confident that local training will not always be needed. Using satellite ET estimates could be a solution, but this would need to be investigated first. Local-scale validation of such products do not always show good results (e.g. Pardo et al., 2014, Cheng et al., 2021). In the future, we aim to train the algorithm using data from a range of sites with different climate and vegetation cover

- What about emissivity? Do the Ts data not need to be corrected for that? And the fact that the measurements are only measured between 8 and 14 micometer(?).

Good point. Because we are focussing on conditions of partial to full vegetation cover NDVI > 0.5, we assumed emissivity effects to be small with typical values close to 0.99. We will mention this in the revised version.

**Specific comments**
Line 24: "Traditionally, ET has been measured through the mass-balance principle applied to catchments or lysimeters". This statement could be expanded a little to help the reader and needs some references.

We will add references to back up this statement, like the reviews by Senay et al. (2011) and Allen et al. (2011).

In fact the whole paragraph between line 24-32 is devoid of any references. This needs fixing.

We will add more references.

LIne 29: Replace "thermal infrared window", by "thermal infrared atmospheric window of the electromagnetic spectrum"

Thanks for the suggestion, will replace.

Line 58: Say: "model CAT S62 Pro; referred to as S62 from hereon)"

Good suggestion, will adopt.

Line 59/60: Explain better what is mean by saturation here, and what causes this? Why these 2 layers of paper, this sounds rather arbitrary.

A standard light sensor on a smartphone will oversaturate when pointed directly at the sun (i.e. sensor output reaches it maximum possible value).  By using a filter such as a small piece of paper, the whole dynamic range of the sensor can be utilized after re-calibration. This procedure is similar to the one proposed by Hukseflux for the Pyranometer App. We will describe this more clearly.

Line 61: Also, what is meant by "phone held straight-up perpendicular to the sun". You mean that the phone is held vertically? Refer to Fig.1a here? Also, why are you taking a photo of obstacles sitting on the surface, rather than of the actual land surface? I find this confusing. Surely, this angle is only suitable/crucial for the operation of the light sensor, not for the IR image?

Correct, we only do this for the light sensor because a standard smartphone lens leads to angle-dependency of the measured light intensity. This intensity is subsequently corrected for a horizontal surface by using the pitch angle as recorded by the smartphone. We found this procedure to work well (Fig 2a). We will describe this more clearly in the revised version.

Line 64 & 65: use subscripts for Ta and ws (and for Ts in line 62). You use them later in the equation.

Thanks for noticing, will change.

Line 70-71: You say "partitioning of incoming solar radiation into evapotranspiration and sensible heat". It is net radiation that is partitioned into evapotranspiration and sensible heat, but also into soil heat flux. So, this statement is incorrect. Also "incoming solar radiation" is the same as "global radiation". Do you want to stick with one term? The latter one is less intuitive.

Good point, this was indeed formulated a bit sloppy. We will use global radiation in a revised version.

Line 71: "Both can be measured by phone's internal sensors.." What does "Both refer to here?

"Both" refers to the incoming radiation and the surface temperature. We will rephrase the sentence to "Both the incoming radiation and the surface temperature can be measured by the phone's internal sensor."

Line 80-81: Can a little bit more information be given here? "The smartphone and Büel observations are available from Teuling and Lammers (2023)". How many measurements/ IR images were taken in the field, and of what kind of surface? Only in the footprint of the EC mast or 'on the lysimeter'?

We will add more information on the measurements and the characteristics of the site.

Line 89: This equation needs a number. Also, the various alphas are not defined properly? Nor is parameter c? Why is pressure not considered? If that is the case ,then take it out of the rest of the paper.

Thanks for the suggestions. We will describe the equation better, and remove pressure from the manuscript.

Line 135-136: What is meant with ".. magnitude of the offset term". Is this the parameter c in the equation?

Correct. Will describe this more clearly.

**Technical corrections**
Line 39: It should be "Hukseflux" not "Hukselfux".
Line 85: It should be "...a lack **of** energy balance closure"
Line 130: 'it should be **negligible** role'.
Line 132: It should be "these condition**s**"

Thanks for spotting these typos. Will be corrected.

References:

Jiang, L., Zhang, B., Han, S., Chen, H. & Wei, Z. (2021), Upscaling evapotranspiration from the instantaneous to the daily time scale: Assessing six methods including an optimized coefficient based on worldwide eddy covariance flux network. Journal of Hydrology, 596, 126135.

Pardo, N., Sánchez, M.L., Timmermans, J., Su, Z., Perez, I.A. & García, M.A. (2014), SEBS validation in a Spanish rotating crop. *Agricultural and forest meteorology*, **195**, 132–142.

Cheng, M., Jiao, X., Li, B., Yu, X., Shao, M. & Jin, X. (2021), Long time series of daily evapotranspiration in China based on the SEBAL model and multisource images and validation. *Earth Syst. Sci. Data*, **13**, 3995–4017, https://doi.org/10.5194/essd-13-3995-2021.

Senay, G.B., Leake, S., Nagler, P.L., Artan, G., Dickinson, J., Cordova, J.T. & Glenn, E.P. (2011), Estimating basin scale evapotranspiration (ET) by water balance and remote sensing methods. *Hydrol. Process.*, **25**, 4037–4049. https://doi.org/10.1002/hyp.8379.

Allen, R.G., Pereira, L.S., Howell, T.A. & Jensen, M.E. (2011), Evapotranspiration information reporting: I. Factors governing measurement accuracy. *Agricultural Water Management*, **98**(6), 899–920.

---

## Author Comment (AC2)

Comments

The main reason behind my decision is that the models proposed by the authors are developed and tested on a "well-instrumented site". What is the guarantee that the proposed model will work elsewhere when there is no lysimeter or EC data available for calibration? What parameters to use? To show the actual feasibility of this approach, the authors should have tested on other sites not used for calibration. What errors can we expect by applying parameters calibrated elsewhere? What are influential factors hindering generalization? Also, how are results affected by the individual mobile phone used?

Our goal with the manuscript is to show that for a particular site, and even with limited training, smartphone-based observations can provide ET estimates that are as accurate as estimates of lysimeter ET by EC and vice versa. We think this (preliminary) finding is important. With this manuscript, we want to invite other groups to start exploring the use of smartphone-based ET estimation so that we can find answers to the questions posed by the reviewer. Our goal is NOT to claim that the model we used should somehow be the basis of all future smartphone ET estimation. We do believe in the general principle of combining smartphone observation with machine learning, but future studies will be able to do so with more training data, and more sophisticated ML methods. This will be made clearer in a revised version. We are currently exploring the accuracy of the initial simple model when applied to other sites. We are happy to include a validation on a different site (likely at one of the TERENO lysimeter sites in Germany) in a revised version as suggested by the reviewer. A full quantification of errors induced by generalization will however not be feasible, as there will likely be important vegetation effects which can not all be investigated with the lysimeters currently available. But it should be noted that other ET estimation methods, such as those based on airborne or spaceborne thermal remote sensing, will suffer from similar if not larger uncertainties that are rarely quantified. Based on our experience, we believe the effect of individual mobile phones is small compared to the uncertainties associated with the Weatherflow sensor. The FLIR Lepton sensor is generally accurate, and within-sensor variability is likely small compared to the averaging/sampling error made when estimating the average surface temperature over the lysimeter or EC footprint. This will also be discussed in more detail in a revised version.

Other issues of relevance include the inappropriate use of the terms "machine learning" to define multivariate regression done with very limited data (machine learning is data hungry), as well as the lack of information concerning the rationale behind the formula used. Other detailed comments follow below.

Multiple linear regression is typically seen as the simplest form of ML, and therefore the best given the limited amount of data we currently have. However it should be noted that the results are robust, as is shown in Fig 4 by the relatively small spread in the distributions of the model parameters after repeating the fitting 2000 times with different samples. The model was chosen because of its simplicity, and because many of the existing ET formulas are (near)linear combinations of global/net radiation, temperature, and humidity. Again, it should be noted that we do believe in the general principle of combining smartphone observation with machine learning, and that future studies will be able to do so with more

training data, and more sophisticated ML methods. This will be made more clear in a revised version.

Lines 24 - 32: Can the authors refer to existing literature when highlighting these gaps?

This was also noted by the other reviewer. We will add more references in a revised version.

Line 44: built-in

Thanks for the correction.

Line 48: not sure referring to the figure without a thorough explanation is suitable at this point in the introduction. Better in the methodological section?

Good point. We will reconsider the referencing to the figure.

Lines 49-50: "[...] the question is how these estimates can work in concert under field conditions to produce accurate ET." this part is not sufficiently clear.

This will be rephrased.

Line 54: RQ1 seems to be very generic; is the focus beyond that of ET?

RQ1 was formulated this way because sufficient accuracy of the individual variables is assumed to be a prerequisite for accurate ET estimation. This will be motivated better in a revised version.

Line 55: RQ2 comes a bit out of the blue. It is the first time machine learning is mentioned. There is no story leading to it.

Thanks for the suggestion. In a revision will be add a paragraph on the use and potential of ML in ET estimation to better link to the RQ.

Figure 1: Not sure why Figure 1 is made of two parts given that they look quite different. Would it make more sense to seprate them? Also, I don't find the caption particularly informative and sufficiently correlated with the images.

We will reconsider the figure and caption of Fig 1.

Line 82: Add a space before "An overview..."

Will correct.

Line 88: "we use the following multivariate regression as a simple form of machine learning to estimate the evapotranspiration..." <- why calling this machine learning? It is a simple multivariate regression, and there is nothing wrong with it, per se. The authors should refrain from calling this machine learning and change the manuscript accordingly. On the other hand, what is the rationale behind the formula used? What are the parameters that need to be calibrated? There is no explanation.

We will use the term multivariate regression when discussing the method that we followed. However we do want to point out that future studies, which likely will have more data available for training, should use more sophisticated ML methods.

Line 89: The equation has not been numbered.

Will correct

Line 93: how much data is available in total?

Indeed this information was not mentioned in the manuscript. All measurements (18 variables on smartphone, fluxes, meteo observations) are available for 36 moments over 4 days (Fig 1B shows some variables over the whole observation period). This will be added to the manuscript.

Figure 2: the legend and captions are confusing, please amend.

Will change.

Line 113: "training Eq. 1" you do not train equation and that equation is not machine learning.

We will use "fit" rather than "train"

Line 117-120: I think the authors overstate the results they obtained. They are fitting their model to the two target variables, EC or lysimeter using values of said variables for parameters calibration (in the "training" dataset). On the other hand, lysimeter and EC are obtained independently.

It is not clear what point the referee aims to make here. We fit the model to EC and lysimeter data independently. These models are then evaluated both against the same method as well as the other (independent) one.

Line 130: I don't this is a great practice to add comments in parentheses? E.g., "it should be noted...".

Good suggestion, will change.

Line 130-133: This paragraph is convoluted. Please rephrase and consider splitting it.

Thanks for the suggestion. Will do.

Figure 3: The caption is not clear and the figure as well. Why not showing with different markers calibration vs validation data?

Good suggestion, will adapt.

Why the two vertical bars separating the two parts of the image?

Thanks for noticing. The bars are probably a result from the merger of the two panels from separate files. This will be fixed.

Discussion and Outlook: I find that this section lacks a proper discussion on the limitation of the proposed approach. How many precision lysimeters do we need to calibrate a world-wide network of phone-based algorithms? Is a linear method sufficient to generalize to location with no calibration data? Perhaps a non-linear machine learning model would be

more useful to improve model generalization by processing external data (i.e., rural/urban catchment characteristics, see [1] for instance.

We agree that the discussion would benefit from discussing these points in more depth. We did not intend to claim that the linear model should be seen as an end-point, but rather as the simplest starting point in a more generic ML-based approach. Future studies that have more data available for training should use more sophisticated ML methods, that allow for the use of categorical variables.

[1] Kratzert, Frederik, et al. "Towards learning universal, regional, and local hydrological behaviors via machine learning applied to large-sample datasets." Hydrology and Earth System Sciences 23.12 (2019): 5089-5110.

---

## Author Response (AR1)

We thank both reviewers and the editor for their constructive comments. It has taken us some time to submit a revision, mainly because additional field testing was requested by the referees. We managed to do so in April 2024 in the German Rollesbroich site. As a result, the manuscript now also discusses the performance in a second site. We have added Belle Holthuis, who performed the field testing, as co-author. We also changed the title to better reflect this is a proof-of-concept. We believe the iteration has significantly improved our manuscript, and we hope for a positive evaluation.

**RC1**

This scientific quality of this note is overall okay, and its contents are moderately interesting and marginally useful for the kind of research or context presented in the paper (micrometeorological or water balance research). It may be useful for irrigation planning or garden water use monitoring, but I am not at all convinced that it marks a "watershed moment in hydrology".

My reservations relate to the fact that:

- generally instantaneous values of weather variables and ET estimates are less useful than hourly values integrated to e.g. daily values. No operator is going to stand in the field for 24/7 to take these measurements.

We fully agree with this point. In many practical applications, one would like an estimate of the daily ET sum. The conversion from instantaneous to daily values was however not our focus, since this is a topic that is already well covered in the literature on satellite estimation of ET where satellite overpasses are typically rare (once a day at best, but more typically once every few days). These methods are equally valid for our approach. In the revised manuscript, we have included a references to Jiang et al. (2021) in which several of the typical upscaling methods are compared. In the Methods section, we included the following: "For many practical applications, the interest would be on daily rather than instantaneous ET values. Various upscaling methods are available from the Earth observation literature to do this (see Jiang et al., 2021), but these are not used in this study since our method is not limited to available satellite overpasses and multiple observations can be taken for a more robust estimation of the daily mean."

- the IR phone images are only useful if one is interested in spatial variability of ET. Perhaps it can be useful for small-scale investigations of ET variability, e.g. in urban settings.

Investigation spatial variability of ET in specific settings including urban environments is indeed a logical application of the method. However we believe there is a wider potential because of the low-cost nature of the sensors. We included the following in the Introduction: "This setup can be used to measure temporal evolution of surface energy balance partitioning at a specific location, or spatial patterns of flux partitioning, in particular in areas with high spatial variability such as urban environments."

- As I said, no operator is going to stand in the field for 24/7 to take these measurements. Why not buy a cheap weather station, e.g. and Ecowitt one, which is barely more expensive

than the WEATHERmeter and supplement it with a cheap surface temperature sensor?

For local applications at smaller scales over which meteorological conditions are not expected to vary significantly, a cheap weather station could indeed provide an alternative to the WEATHERmeter. When supplemented by surface temperature sensor, the same methodology could in principle be applied to the resulting data. However this would still require post-processing on a separate computer. The advantage of using a smartphone as platform is that with a dedicated app, all the flux properties can be calculated, shown, and used on the fly. This is now mentioned in the Discussion: "The same principle used here on smartphone data could potentially also be applied to a combination of a cheap weather station and IR temperature sensor for a more automated monitoring at a single location. However this would require post-processing on the computer, while a dedicated smartphone app could do the same on the fly."

- In fact, how well would the model have done with Ta only and not Ts? Do we really need Ts? Can this be tested and discussed?

Ultimately, both sensible and latent heat fluxes are to a large extent driven by the land-atmosphere temperature gradient. This is illustrated by Fig3a and Fig 5. Fig 3a shows the relation between the temperature difference Ts-Ta and the sensible heat flux. Since most of the net radiation that is not used for sensible heat is used for ET, this already shows that the temperature difference, and not Ta, is the main driving force of ET. Fig 5 further investigates the importance of the different variables. As can be seen, both Ta and Ts carry significant information on ET (i.e. contribution differs from 0) , so using only Ta would not result in robust estimates. It should be noted that observations were made in humid (non-water-limited) conditions. Under water-limited conditions, the role of Ts will likely be more important. This will be tested in the future. We also added the following in reference to Fig 2: "It also illustrates the temporal dynamics of the difference between surface and air temperature, which is largest near the daily global radiation peak reflecting the strongest turbulent heat fluxes" to make the role of temperature in driving heat fluxes more clear.

- This approach is only useful if there are high-quality ET data (e.g. EC-data) available to calibrate the ML model. For most places and users these are not available, and I am not sure how we would get around that. Unless we used cloud free high-resolution ET estimates to calibrate the ML models for specific settings, where lots of hobbyist weather station data are available.

To account for this comment and a similar comment by the other referee on extrapolation of the results to sites without ET observations for calibration, we have made 2 important changes. First, we have made more clear that this is a first feasibility study, and that the specific model used should be replaced by more sophisticated ML methods once more data from more sites become available for training/calibration. Also, we have changed the title into: "Investigating the potential for smartphone-based monitoring of evapotranspiration and land surface energy balance partitioning" to indicate this is feasibility study. Second, as part of the revision we performed a second field campaign with the same instruments at the Rollesbroich catchment in Germany. This has provided valuable new insights on the calibration issue, as well as clear recommendations for further study. We have made this

new dataset available on hydroshare, and discuss the results in the Discussion section where we have also added an additional figure on this analysis:

"…While these results show that smartphone ET estimation can give accurate values after local calibration, they do not provide information on the performance at other sites where no calibration data is available.

In order to investigate the transferability of the method to other sites, a second measurement campaign was conducted in April 2024 at the TERENO lysimeters located in the Rollesbroich hydrological observatory (Qu et al., 2016). Direct application of the model (Eq. 1) calibrated to subsets of the Büel observations as described earlier gave a satisfactory model performance with a median RMSE of around 0.10 mm/h for each of the six lysimeters. This performance however increased considerably after local calibration following the same procedure as used earlier for Büel, with median RMSE values in the range 0.06–0.07 mm/h. This shows that the general methodology works at different sites but best results are obtained after calibration. A closer look into the difference between the models calibrated on Büel vs. Rollesbroich data (Figure 6) provides an explanation for the poorer model performance. Besides warmer temperatures encountered at Büel, wind conditions were lower during the Rietholzbach campaign (order 0.5–1 m/s) than during the Rollesbroich campaign (4–5 m/s). As a result, the gradient between surface and air temperature was much smaller at Rollesbroich, and wind becomes a more important predictor in the model (Figure 6) at the expense of temperature. This shows that for future application, a more complex model that is trained on a more complete range of weather conditions is needed."

- What about emissivity? Do the Ts data not need to be corrected for that? And the fact that the measurements are only measured between 8 and 14 micometer(?).

Good point. Because we are focussing on conditions of partial to full vegetation cover NDVI > 0.5, we assumed emissivity effects to be small with typical values close to 0.99. This is now mentioned in the Methods: "Because we focus on vegetated conditions, we assume emissivity does not differ from unity."

**Specific comments**
Line 24: "Traditionally, ET has been measured through the mass-balance principle applied to catchments or lysimeters". This statement could be expanded a little to help the reader and needs some references.

We have added references to back up this statement, like the reviews by Senay et al. (2011) and Allen et al. (2011).

In fact the whole paragraph between line 24-32 is devoid of any references. This needs fixing.

We added more references.

LIne 29: Replace "thermal infrared window", by "thermal infrared atmospheric window of the electromagnetic spectrum"

Thanks for the suggestion, will replace.

Line 58: Say: "model CAT S62 Pro; referred to as S62 from hereon)"

Good suggestion, will adopt.

Line 59/60: Explain better what is mean by saturation here, and what causes this? Why these 2 layers of paper, this sounds rather arbitrary.

A standard light sensor on a smartphone will oversaturate when pointed directly at the sun (i.e. sensor output reaches it maximum possible value).  By using a filter such as a small piece of paper, the whole dynamic range of the sensor can be utilized after re-calibration. This procedure is similar to the one proposed by Hukseflux for the Pyranometer App. This is now described more clearly: " in order to avoid sensor saturation when the sensor is exposed to direct sunlight"

Line 61: Also, what is meant by "phone held straight-up perpendicular to the sun". You mean that the phone is held vertically? Refer to Fig.1a here? Also, why are you taking a photo of obstacles sitting on the surface, rather than of the actual land surface? I find this confusing. Surely, this angle is only suitable/crucial for the operation of the light sensor, not for the IR image?

Correct, we only do this for the light sensor because a standard smartphone lens leads to angle-dependency of the measured light intensity. This intensity is subsequently corrected for a horizontal surface by using the pitch angle as recorded by the smartphone. We found this procedure to work well (Fig 2a). This is described more clearly in the revised version: "luminance (I) measurements were taken with the phone held straight-up perpendicular to the sun".

Line 64 & 65: use subscripts for Ta and ws (and for Ts in line 62). You use them later in the equation.

Thanks for noticing, has been changed.

Line 70-71: You say "partitioning of incoming solar radiation into evapotranspiration and sensible heat". It is net radiation that is partitioned into evapotranspiration and sensible heat, but also into soil heat flux. So, this statement is incorrect. Also "incoming solar radiation" is the same as "global radiation". Do you want to stick with one term? The latter one is less intuitive.

Good point, this was indeed formulated a bit sloppy. We will use global radiation in a revised version.

Line 71: "Both can be measured by phone's internal sensors.." What does "Both refer to here?

"Both" refers to the incoming radiation and the surface temperature. We will rephrase the sentence to "Both the global radiation and the surface temperature can be measured by the phone's internal sensor."

Line 80-81: Can a little bit more information be given here? "The smartphone and Büel observations are available from Teuling and Lammers (2023)". How many measurements/ IR images were taken in the field, and of what kind of surface? Only in the footprint of the EC mast or 'on the lysimeter'?

We will add more information on the measurements and the characteristics of the site.

Line 89: This equation needs a number. Also, the various alphas are not defined properly? Nor is parameter c? Why is pressure not considered? If that is the case ,then take it out of the rest of the paper.

Thanks for the suggestions. We will describe the equation better, and remove pressure from the manuscript.

Line 135-136: What is meant with ".. magnitude of the offset term". Is this the parameter c in the equation?

Correct. We added "intercept $c$" in the revision for clarity.

**Technical corrections**
Line 39: It should be "Hukseflux" not "Hukselfux".
Line 85: It should be "...a lack **of** energy balance closure"
Line 130: 'it should be **negligible** role'.
Line 132: It should be "these condition**s**"

Thanks for spotting these typos. They have been corrected.

References:

Jiang, L., Zhang, B., Han, S., Chen, H. & Wei, Z. (2021), Upscaling evapotranspiration from the instantaneous to the daily time scale: Assessing six methods including an optimized coefficient based on worldwide eddy covariance flux network. Journal of Hydrology, 596, 126135.

Pardo, N., Sánchez, M.L., Timmermans, J., Su, Z., Perez, I.A. & García, M.A. (2014), SEBS validation in a Spanish rotating crop. *Agricultural and forest meteorology*, **195**, 132–142.

Cheng, M., Jiao, X., Li, B., Yu, X., Shao, M. & Jin, X. (2021), Long time series of daily evapotranspiration in China based on the SEBAL model and multisource images and validation. *Earth Syst. Sci. Data*, **13**, 3995–4017, https://doi.org/10.5194/essd-13-3995-2021.

Senay, G.B., Leake, S., Nagler, P.L., Artan, G., Dickinson, J., Cordova, J.T. & Glenn, E.P. (2011), Estimating basin scale evapotranspiration (ET) by water balance and remote sensing methods. *Hydrol. Process.*, **25**, 4037–4049. https://doi.org/10.1002/hyp.8379.

Allen, R.G., Pereira, L.S., Howell, T.A. & Jensen, M.E. (2011), Evapotranspiration information reporting: I. Factors governing measurement accuracy. *Agricultural Water Management*, **98**(6), 899–920.

**RC2**

The main reason behind my decision is that the models proposed by the authors are developed and tested on a "well-instrumented site". What is the guarantee that the proposed model will work elsewhere when there is no lysimeter or EC data available for calibration? What parameters to use? To show the actual feasibility of this approach, the authors should have tested on other sites not used for calibration. What errors can we expect by applying parameters calibrated elsewhere? What are influential factors hindering generalization? Also, how are results affected by the individual mobile phone used?

To account for this comment and a similar comment by the other referee on extrapolation of the results to sites without ET observations for calibration, we have made 2 important changes. First, we have made more clear that this is a first feasibility study, and that the specific model used should be replaced by mor sophisticated ML methods once more data from more sites become available. Also, we have changed the title into: "Investigating the potential for smartphone-based monitoring of evapotranspiration and land surface energy balance partitioning" to indicate this is feasibility study. Second, as part of the revision we performed a second field campaign with the same instruments at the Rollesbroich catchment in Germany. This has provided valuable new insights, as well as clear recommendations for further study. We made the new dataset available on hydroshare, and discuss the results in the Discussion section where we have also added an additional figure on this analysis. We believe most aspects are mentioned now in the discussion, including a discussion of the ET errors at Rollesbroich, but we should stress that a full quantification of all errors and uncertainties under a range of conditions is illusive at this point and will require several additional field campaigns.

Other issues of relevance include the inappropriate use of the terms "machine learning" to define multivariate regression done with very limited data (machine learning is data hungry), as well as the lack of information concerning the rationale behind the formula used. Other detailed comments follow below.

Multiple linear regression is typically seen as the simplest form of ML, and therefore the best given the limited amount of data we currently have. However it should be noted that the results are robust, as is shown in Fig 4 by the relatively small spread in the distributions of the model parameters after repeating the fitting 2000 times with different samples. The model was chosen because of its simplicity, and because many of the existing ET formulas are (near)linear combinations of global/net radiation, temperature, and humidity. Again, it should be noted that we do believe in the general principle of combining smartphone observation with machine learning, and that future studies will be able to do so with more training data, and more sophisticated ML methods. In the revised version, this is now explained more clearly, and we also refrain from using ML in the manuscript following feedback by the other referee.

Lines 24 - 32: Can the authors refer to existing literature when highlighting these gaps?

This was also noted by the other reviewer. We have added more references in the revised version.

Line 44: built-in

Thanks for the correction.

Line 48: not sure referring to the figure without a thorough explanation is suitable at this point in the introduction. Better in the methodological section?

Good point. We now reference the figure at the beginning of the Methods section, and have split the figure in response to another suggestions by the same referee.

Lines 49-50: "[...] the question is how these estimates can work in concert under field conditions to produce accurate ET." this part is not sufficiently clear.

This has been rephrased into "While smartphones can potentially monitor all variables relevant for ET, the question is if these estimates, when combined, provide enough information for accurate ET estimation under field conditions."

Line 54: RQ1 seems to be very generic; is the focus beyond that of ET?

RQ1 was re-formulated into "Do handheld sensors provide robust estimates of standard meteorological variables relevant for ET estimation?".

Line 55: RQ2 comes a bit out of the blue. It is the first time machine learning is mentioned. There is no story leading to it.

Thanks for the suggestion. We have now added some text in the Intro on ET estimation to better link to the RQ: "While such approaches give valuable insight into the spatial distribution of ET, they rely on available satellite overpasses, and cloud-free conditions, and ET inference models (Amani and Shafizadeh-Moghadam, 2023). Most ET inference models and more classical potential ET-based methods have been developed in times when actual ET observations were scarce. Due to increasing availability of observations in hydrology, but also ET in particular, machine learning approaches now often outperform existing models due to their ability to optimally utilize the information in observations (Kratzert et al., 2019)."

Figure 1: Not sure why Figure 1 is made of two parts given that they look quite different. Would it make more sense to seprate them? Also, I don't find the caption particularly informative and sufficiently correlated with the images.

We have split Fig 1 and revised the caption.

Line 82: Add a space before "An overview..."

Done.

Line 88: "we use the following multivariate regression as a simple form of machine learning to estimate the evapotranspiration..." <- why calling this machine learning? It is a simple multivariate regression, and there is nothing wrong with it, per se. The authors should refrain from calling this machine learning and change the manuscript accordingly. On the other hand, what is the rationale behind the formula used? What are the parameters that need to be calibrated? There is no explanation.

We now use the term multivariate regression when discussing the method that we followed. However we do want to point out that future studies, which likely will have more data available for training, should use more sophisticated ML methods.

Line 89: The equation has not been numbered.

Corrected.

Line 93: how much data is available in total?

Indeed this information was not mentioned in the manuscript. All measurements (18 variables on smartphone, fluxes, meteo observations) are available for 36 moments over 4 days (Fig 1B shows some variables over the whole observation period). This has been added to the Method section and the caption of the new Figure 2.

Figure 2: the legend and captions are confusing, please amend.

We have added additional explanation to the caption.

Line 113: "training Eq. 1" you do not train equation and that equation is not machine learning.

We now use "fit" rather than "train"

Line 117-120: I think the authors overstate the results they obtained. They are fitting their model to the two target variables, EC or lysimeter using values of said variables for parameters calibration (in the "training" dataset). On the other hand, lysimeter and EC are obtained independently.

Indeed the independency of EC and lysimeter data, and having those available at a single site, was the main reason we selected Rietholzbach for our first field campaign. This is now made more clear in the Methods and Data section: "The site was chosen because ET is measured independently by a large weighing lysimeter and eddy covariance."

Line 130: I don't this is a great practice to add comments in parentheses? E.g., "it should be noted...".

Good suggestion, changed.

Line 130-133: This paragraph is convoluted. Please rephrase and consider splitting it.

Thanks for the suggestion. Several changes were made to this paragraph in the revision.

Figure 3: The caption is not clear and the figure as well. Why not showing with different markers calibration vs validation data?

Good suggestion, will adapt. The caption has been made more clear by adding: "(for lysimeter and eddy covariance)".

Why the two vertical bars separating the two parts of the image?

Thanks for noticing. The bars are probably a result from the merger of the two panels from separate files. This has now been fixed.

Discussion and Outlook: I find that this section lacks a proper discussion on the limitation of the proposed approach. How many precision lysimeters do we need to calibrate a world-wide network of phone-based algorithms? Is a linear method sufficient to generalize to location with no calibration data? Perhaps a non-linear machine learning model would be more useful to improve model generalization by processing external data (i.e., rural/urban catchment characteristics, see [1] for instance.

We have added analysis of a second dataset to the Discussion, and we emphasize that future research should use a more complex/nonlinear NL model trained on more data when estimating ET at a site without ET observations for calibration.

[1] Kratzert, Frederik, et al. "Towards learning universal, regional, and local hydrological behaviors via machine learning applied to large-sample datasets." Hydrology and Earth System Sciences 23.12 (2019): 5089-5110.